# In Silico Design of Quantitative Polymerase Chain Reaction (qPCR) Assay Probes for Prostate Cancer Diagnosis, Prognosis, and Personalised Treatment

**DOI:** 10.3390/cimb47040292

**Published:** 2025-04-19

**Authors:** Trevor Kenneth Wilson, Oliver Tendayi Zishiri

**Affiliations:** Discipline of Genetics, College of Agriculture, Engineering and Science, University of KwaZulu-Natal, Durban 4000, South Africa; 218014013@stu.ukzn.ac.za

**Keywords:** prostate cancer, diagnosis, prognosis, personalised treatment, genetic mutations, qPCR assay

## Abstract

Prostate cancer is one of the world’s leading causes of cancer-related mortalities. There are several diagnostic tools and treatment plans readily available, such as prostate-specific antigen (PSA) tests and androgen deprivation therapy (ADT). However, these all come with their setbacks. Therefore, alternatives must be developed to assist those patients for whom standardised treatment does not work. There are many genes whose mutations lead to prostate cancer development and progression. These mutations may also lead to higher resistance/vulnerability to specific therapies. In this in silico study, four genes, AR, ATM, PTEN, and TP53, were assessed, and mutations were chosen for qPCR primer and probe design. A total of 28 mutations were selected from the four genes, with PTEN (13) making up the majority of the mutations, followed by TP53 (six), then ATM (five), and finally, AR (four). All primer/probe combinations fall within the desired ranges for this study and provide valuable additions to prostate cancer’s diagnostic/prognostic landscape. These assays will require further experimental validation, but they are the first step toward a better future in the fight against this horrible disease.

## 1. Introduction

Prostate cancer is the second most common cancer in men worldwide [1]. There are several risk factors associated with prostate cancer, including age, ethnicity, obesity, and familial risk through inherited gene alterations [2]. Standard diagnostic tools currently in use for prostate cancer include digital rectal examinations, prostate-specific antigen (PSA) tests, and prostate biopsies [3]. However, these tests can be invasive and harbour risks of false-positive diagnoses [4]. Currently, the most common treatment options for prostate cancer are Androgen deprivation therapies, radical prostatectomy, or radiation [2]. Many men often experience relapsed prostate cancer after treatment, with more aggressive forms such as metastatic castration-resistant prostate cancer (mCRPC) being common [4]. An accurate depiction of prostate cancer in Africa is not available due to minimal testing being conducted. However, from the known data, prostate cancer is the most common cause of cancer-related deaths among African men, with as many as 38 in 100,000 men having this form of cancer and with as high as 64% of new cases showing advanced stages of the disease [5].

Gene alterations are found regularly in prostate cancer, with several genes playing a crucial role in this disease. One such gene is the Androgen receptor (AR) gene, encoding the Androgen receptor, a transcription factor dependent on binding ligands such as testosterone to itself [6]. AR signalling plays key roles in regular prostate development and, in prostate cancer, is upregulated [7]. Therefore, the mainstay therapy for prostate cancer tends to be Androgen deprivation therapy (ADT) through antiandrogens and AR inhibitors such as enzalutamide [6,8]. However, specific AR mutations may lead to resistances, or susceptibilities, to different ADTs, so it is essential to determine which therapies will work for individual patients.

Where AR upregulation is found in prostate cancer, the ATM (ataxia telangiectasia mutated) gene’s downregulation, or complete loss, is found [9,10]. The protein kinase ATM is involved in DNA damage responses [11]. However, this gene’s loss is also associated with the development of its namesake neurodegenerative disease, Ataxia Telangiectasia (AT), along with subsets of prostate cancer [12]. There are treatment methods for this subset but the identification of mutations of this gene within prostate cancer patients is necessary to determine whether these treatments will work.

Another gene whose loss is heavily involved in prostate cancer development is the phosphatase and tensin homologue (PTEN) gene. The PTEN gene encodes the PTEN protein, which plays a crucial role in the phosphatidylinositol-3-OH kinase (PI3K), protein kinase B (AKT), mammalian target of rapamycin (mTOR) signalling pathway [13]. Loss-of-function mutations in the PTEN gene lead to the overexpression of this signalling pathway, causing uncontrolled cell growth and proliferation, leading to the development and progression of prostate cancer [14]. PTEN loss is found more commonly in patients with advanced prostate cancer compared to those with the primary form of disease [15]. PTEN restoration, PI3K inhibition, and AKT inhibition are all techniques to combat prostate cancer [16,17,18]. Therefore, PTEN loss can be seen as both an indicator of advanced prostate cancer as well as a marker for specific treatment options, similar to the next gene of interest, TP53.

The tumour protein p53 (TP53) gene is often found mutated in multiple cancers, breast cancer and prostate cancer, included [19]. TP53 mutations are most likely to be found in late-stage mCRPC, but evidence has shown a reasonably high mutation frequency of TP53 in the primary stages of the disease [20,21,22]. With this, TP53-deficient prostate cancers have been shown to improve when treated with gambogic acid through inhibition of the mitogen-activated protein kinase (MAPK) pathway [23]. TP53 mutations could be used to identify whether specific treatments will work for individual patients.

This study aims to use mutational data to develop primers and probes for qPCR assays that are intended to help determine personalised treatment plans for men with prostate cancer while also being used for diagnosis and prognosis of patients. These assays will be developed in silico and require further experimental validation.

## 2. Methods

### 2.1. Mutational Data

Mutations were chosen from four different genes commonly associated with prostate cancer development and progression, namely the Androgen receptor, Ataxia-telangiectasia mutated, Phosphatase and tensin homologue, and tumour protein 53 genes (AR, ATM, PTEN, and TP53, respectively). The mutational data were compiled, which included the type of mutation, whether it be an SNP, indel, or structural variant, and the chromosomal location of the mutation. The mutations were chosen from various criteria: the predicted effect of the mutation (stop-gained, frameshift, modifying effect, etc.), the predicted impact (high or moderate), the predicted effect on protein function (deleterious), and its clinical significance (benign or pathogenic) (Figure 1).

### 2.2. Primer Design

PrimerQuest™ (Integrated DNA Technologies, Coralville, IA, USA) was used to design the primers for the qPCR reaction. The primers were to be ideally within 100–200 bases of the mutation site for optimal qPCR performance. However, due to some variants’ flanking regions being undesirable for primer design, leeway was given to allow up to 400 bp amplicon length. Primer melting temperatures (Tms) were between 58 and 65 °C, with an optimal Tm between 60 and 62 °C. Primer GC contents were to be strictly between 40 and 60%. Tms were determined using the following parameters: a monovalent salt concentration of 50 mM, a divalent salt concentration of 1.5 mM, a primer concentration of 400 nM, a probe concentration of 250 nM, and a dNTP concentration of 0.6 mM.

### 2.3. Probe Design

Firstly, the target region for the probe was defined as the 20–30 bp region flanking the mutation. The probe length was restricted to 18–30 bp with a GC content of between 40 and 60% and Tm of 65–75 °C, roughly 5–10 °C above the Tm of the primers, to ensure optimal conditions and stable binding. PrimerQuest™ was, again, used to design the probes. Probes for both the wild type and mutant alleles were designed, each with a different dyes to distinguish between the two, to allow for a more controlled test result. For all wild-type probes, 6-Carboxyfluorescein (6-FAM) was used as the fluorophore, with Black Hole Quencher 1 (BHQ1) as the quencher, while mutant probes used hexachloro-fluorescein (HEX) as the fluorophore, with BHQ1 as the quencher. The PrimerQuest™ settings remain the same here as those presented in the Primer Design Section.

### 2.4. In Silico Probe Validation

To avoid potential cross-hybridisation, BLAST was used to check for similarities to any other unintended regions of the genome and ensure the uniqueness of the primers and probes to the target site. The core nucleotide database was used for the search, with the organism set as homo sapiens (taxid 9606) and the programme optimised for highly similar sequences. All other BLAST parameters remained as the default. Thereafter, the UCSC In Silico PCR [24] was used to ensure that the primer pairs did not amplify non-target regions across the human genome. The Thermo Fisher Scientific Multiple Primer Analyzer was used to check for self and cross-dimerisation amongst primers and probes. The settings for the Multiple Primer Analyzer were set to match those used in PrimerQuest™, with the value for dimerisation set to three. If either form of dimerisation was found present, the primers/probes involved were redesigned to ensure no dimerisation. Using IDT’s OligoAnalyzer, the primers and probes were checked for secondary structures and hairpin formation.

## 3. Results

### 3.1. Mutational Data

FFrom the four aforementioned genes, twenty-eight mutations were selected based on the criteria previously alluded to (Table 1). Four mutations came from the AR gene(Table 2), five from the ATM gene (Table 3), thirteen from the PTEN gene (Table 4), and six from the TP53 gene (Table 5). The mutations included missense, stop-gained, frameshift, and splice region variants. Each of the consequences shows possible alterations to the encoded proteins, which can have devastating effects on the prostate.

### 3.2. qPCR Primers

The four mutations chosen for the AR gene each have primers that amplify regions ranging from 84 to 106 bp in length (Table 2). The AR2 mutation required two sets of primers to be designed, one for the wild-type allele and one for the mutant allele. This was due to the nature of the mutation, making it necessary to design separate primer pairs for the two sequences. Each set of primers had a Tm of 62 °C, with primer lengths ranging from 19 to 24 bp and GC contents ranging from 43.5 to 52.6%.

**Table 2 cimb-47-00292-t002:** Primers and probes designed to amplify and identify the selected AR gene mutations.

ID	OLIGO	Length	Tm	GC%	Sequence	Amplicon Length	BLAST e-Value	Hairpin Tm	Hairpin ΔG
AR1	Forward Primer	24	62	45.8	TCTCCCTCTTATTGTTCCCTACAG	84	2.00 × 10^−4^	29.6	−0.33
	Reverse Primer	22	62	45.5	GCTCACCATGTGTGACTTGATT		0.002	52.4	−2.14
	Wild-Type Probe	26	68	50	TGCGAGAGAGCTGCATCAGTTCACTT		1.00 × 10^−5^	42.3	−2.06
	Mutant Probe	24	69	58	TGCGAGAGAGCTGCATCAGTTCGC		2.00 × 10^−4^	43.8	−2.95
AR2 Wild-Type	Forward Primer	19	62	52.6	ACCTCCTTGTCAACCCTGT	106	0.055	17.2	0.48
	Reverse Primer	23	62	43.5	GCTCACCATGTGTGACTTGATTA		6.00 × 10^−4^	52.4	−2.14
	Wild-Type Probe	30	70	50	TTGTTCCCTACAGATTGCGAGAGAGCTGCA		8.00 × 10^−8^	33.4	−1.27
AR2 Mutant	Forward Primer	24	62	45.8	TCTCCCTCTTATTGTTCCCTACAG	84	2.00 × 10^−4^	29.6	−0.33
	Reverse Primer	22	62	45.5	GCTCACCATGTGTGACTTGATT		2.00 × 10^−3^	52.4	−2.14
	Mutant Probe	30	68	43	AGAGCTGTATCAGTTCACTTTTGACCTGCT		2.00 × 10^−5^	42.3	−2.06
AR3	Forward Primer	21	62	52.4	CATTGAGCCAGGTGTAGTGTG	100	5.00 × 10^−3^	7.2	1.18
	Reverse Primer	20	62	50	AAGCTGTCTCTCTCCCAGTT		2.10 × 10^−2^	31.7	−0.62
	Wild-Type Probe	30	70	53	CCTTTGCAGCCTTGCTCTCTAGCCTCAATG		8.00 × 10^−8^	49.7	−2.39
	Mutant Probe	30	70	53	CCTTTGCAGCCTTGCACTCTAGCCTCAATG		2.00 × 10^−5^	52.6	−2.71
AR4	Forward Primer	21	62	52.4	GACCAGATGGCTGTCATTCAG	98	5.00 × 10^−3^	43.4	−1.6
	Reverse Primer	22	62	45.5	AAGTAGAGCATCCTGGAGTTGA		2.00 × 10^−3^	20.4	0.21
	Wild-Type Probe	24	69	54	ATGGGCTGGCGATCCTTCACCAAT		2.00 × 10^−4^	33.8	−0.61
	Mutant Probe	24	69	54	TGGTGTTTGCCATGGGCTAGCGAT		4.20 × 10^−2^	50	−2.16

Four of the five ATM mutations chosen for primer design only required one set of primers for both the wild-type and mutant alleles, while the ATM5 mutation required separate primer pairs for the wild-type and mutant alleles (Table 3). Primer Tms ranged from 57 to 62 °C, while GC content for primers ranged from 40 to 52.6%. Primer lengths ranged from 19 to 25 bp, with amplicon lengths from 163 to 265 bp.

**Table 3 cimb-47-00292-t003:** Primers and probes designed to amplify and identify the selected ATM gene mutations.

ID	OLIGO	Length	Tm	GC%	Sequence	Amplicon Length	BLAST e-Value	Hairpin ΔG	Hairpin Tm
ATM1	Forward Primer	22	60	40.9	GTTGCTTGGTTCTTTGTTTGTC	265	2.00 × 10^−3^	1.86	−8.2
	Reverse Primer	19	60	52.6	CGCACCTGGCCTTAATTTC		5.50 × 10^−2^	−0.84	38.5
	Wild-Type Probe	29	68	45	AGATTGCATCTGGCTTTTTCCTGCGATTG		3.00 × 10^−7^	−1.44	45
	Mutant Probe	29	69	48	AGATTGCATCTGGCTTTTTCCTGCGATGG		5.00 × 10^−6^	−1.6	39.6
ATM2	Forward Primer	21	57	42.9	GAGTTGGGAGTTACATATTGG	163	0.005	0.41	17.7
	Reverse Primer	20	59	45	CACCACAGCCATACAAACTA		0.021	1.17	−11.2
	Wild-Type Probe	30	64	43	CTTAGAAATCTACAGAAGTATAGGGGAGCC		8.00 × 10^−8^	−0.64	30.8
	Mutant Probe	30	64	43	CCTTAGAAATCTACAGAAGTATAGGGGAGC		3.00 × 10^−7^	−0.34	31.2
ATM3	Forward Primer	25	60	40	GAACTTCTGAAACCACTATCGTAAG	198	5.00 × 10^−5^	0.54	18.7
	Reverse Primer	25	61	40	GGTTTTTCACTACATGAAGGACATG		5.00 × 10^−5^	−3.1	52.6
	Wild-Type Probe	30	68	47	CCACAGCAATGTGTGTTCTTTGTATCGTCG		8.00 × 10^−8^	−4.49	64.4
	Mutant Probe	30	66	43	CCACAACAATGTGTGTTCTTTGTATCGTCG		2.00 × 10^−5^	−3.01	51.1
ATM4	Forward Primer	22	59	40.9	GGACTCAACATAGGCTTAATGA	217	2.00 × 10^−3^	−0.96	36.1
	Reverse Primer	21	59	47.6	CCAGAGAAATCCAGAGGAAAG		5.00 × 10^−3^	−0.7	36.9
	Wild-Type Probe	30	68	47	ATATCCATCATCCGAAAGGAGCCAAAACCC		8.00 × 10^−8^	−2.6	56.5
	Mutant Probe	30	67	43	ATATCCATCATCTGAAAGGAGCCAAAACCC		2.00 × 10^−5^	−1.33	34.9
ATM5 Wild Type	Forward Primer	24	62	41.7	CTCAGACTGACGGATTAAATTCCA	198	2.00 × 10^−4^	−1.21	38.2
	Reverse Primer	25	60	40	GGAATCCACTAGTTCTGTTATGATG		5.00 × 10^−5^	−0.14	26.4
	Wild-Type Probe	30	69	47	TCCAAGGCTATTCAGTGTGCGAGGTAATCT		8.00 × 10^−8^	−0.11	25.9
ATM5 Mutant	Forward Primer	22	60	45.5	GTTACCAAAGGATGCTGTTCTC	243	2.00 × 10^−3^	−0.47	33.7
	Reverse Primer	21	60	42.9	GAAAGTGTTGGACTTGGTTGT		5.00 × 10^−3^	−1.91	49.9
	Mutant Probe	28	66	43	TAAGGCTATTCAGTGTGCGAGGTAATCT		5.00 × 10^−6^	0.09	23.8

**Table 4 cimb-47-00292-t004:** Primers and probes designed to amplify and identify the selected PTEN gene mutations.

ID	OLIGO	Length	Tm	GC%	Sequence	Amplicon Length	BLAST e-Value	Hairpin ΔG	Hairpin Tm
PTEN1	Forward Primer	23	62	43.5	TGACCAATGGCTAAGTGAAGATG	133	6.00 × 10^−4^	0.16	22.4
	Reverse Primer	20	62	50	TAGGGCCTCTTGTGCCTTTA		2.10 × 10^−2^	−2.07	47.6
	Wild-Type Probe	30	68	43	AGGGACGAACTGGTGTAATGATATGTGCAT		8.00 × 10^−8^	0.13	21.9
	Mutant Probe	30	69	47	AGGGACGAACTGGTGTAAGGATATGTGCAT		2.00 × 10^−5^	0.13	21.9
PTEN2	Forward Primer	23	60	43.5	CTCTGTATTAGTGGCATCACAAGT	336	2.00 × 10^−4^	−2.13	47.6
	Reverse Primer	23	60	47	AGTACATTCATACCTACCTCTGC		6.00 × 10^−4^	0.11	23.7
	Wild-Type Probe	30	69	47	AGTTGTGCTGAAAGACATTATGACACCGCC		8.00 × 10^−8^	−0.51	36.5
	Mutant Probe	30	68	43	AATTGTGCTGAAAGACATTATGACACCGCC		1.00 × 10^−6^	−0.51	36.5
PTEN3	Forward Primer	20	60	50	CATAACCCACCACAGCTAGA	261	2.10 × 10^−2^	2.89	−68.7
	Reverse Primer	21	60	47.6	TCAGATCCAGGAAGAGGAAAG		0.005	−0.61	34.3
	Wild-Type Probe	26	68	50	AAAGCTGGAAAGGGACGAACTGGTGT		1.00 × 10^−5^	−1	40.9
	Mutant Probe	25	69	52	AAGCTGGAAACGGACGAACTGGTGT		0.012	−1.17	44.3
PTEN4	Forward Primer	22	63	50	GTTTGTGGTCTGCCAGCTAAAG	97	0.002	−1.17	40.7
	Reverse Primer	21	63	52.4	CGGCTGAGGGAACTCAAAGTA		0.005	−3.18	58
	Wild-Type Probe	24	69	58	TTCAGGACCCACACGACGGGAAGA		2.00 × 10^−4^	−2.43	55.3
	Mutant Probe	25	69	56	TCAGGACCCACATGACGGGAAGACA		0.012	−2.43	55.3
PTEN5	Forward Primer	22	60	45.5	GACCAATGGCTAAGTGAAGATG	164	2.00 × 10^−3^	0.16	22.4
	Reverse Primer	21	60	47.6	TTGTCTCTGGTCCTTACTTCC		0.005	0.06	24.1
	Wild-Type Probe	26	68	50	AAAGCTGGAAAGGGACGAACTGGTGT		1.00 × 10^−5^	−1	40.9
	Mutant Probe	30	68	43	AGCTGGAAAGGGATGAACTGGTGTAATGAT		8.00 × 10^−8^	−1	40.9
PTEN6	Forward Primer	22	60	45.5	GACCAATGGCTAAGTGAAGATG	164	2.00 × 10^−3^	0.16	22.4
	Reverse Primer	21	60	47.6	TTGTCTCTGGTCCTTACTTCC		5.00 × 10^−3^	0.06	24.1
	Wild-Type Probe	26	68	50	AAAGCTGGAAAGGGACGAACTGGTGT		1.00 × 10^−5^	−1	40.9
	Mutant Probe	30	68	43	AGCTGGAAAGGGACAAACTGGTGTAATGAT		8.00 × 10^−8^	−1.66	34.4
PTEN7 Wild Type	Forward Primer	21	62	47.6	TTTCTGTCCACCAGGGAGTAA	148	5.00 × 10^−3^	−1.47	42.4
	Reverse Primer	21	62	47.6	CCGCCACTGAACATTGGAATA		5.00 × 10^−3^	−1.29	41.3
	Wild-Type Probe	24	68	54	TTCCCAGTCAGAGGCGCTATGTGT		2.00 × 10^−4^	0.22	22.2
PTEN7 Mutant	Forward Primer	24	62	41.7	CGACCCAGTTACCATAGCAATTTA	155	2.00 × 10^−4^	1.24	5.8
	Reverse Primer	25	62	40	TCCAGATGATTCTTTAACAGGTAGC		5.00 × 10^−5^	−0.06	26.4
	Mutant Probe	30	69	53	CAGGGAGTAACTATTCCCAGTCAGAGGTGC		5.00 × 10^−6^	−4.14	52.9
PTEN8 Wild Type	Forward Primer	21	62	47.6	TTTCTGTCCACCAGGGAGTAA	148	5.00 × 10^−3^	−1.47	42.4
	Reverse Primer	21	62	47.6	CCGCCACTGAACATTGGAATA		5.00 × 10^−3^	−1.29	41.3
	Wild-Type Probe	24	68	54	TTCCCAGTCAGAGGCGCTATGTGT		2.00 × 10^−4^	0.22	22.2
PTEN8 Mutant	Forward Primer	24	62	41.7	CGACCCAGTTACCATAGCAATTTA	155	2.00 × 10^−4^	1.24	5.8
	Reverse Primer	25	62	40	TCCAGATGATTCTTTAACAGGTAGC		5.00 × 10^−5^	−0.06	26.4
	Mutant Probe	30	69	50	AGGGAGTAACTATTCCCAGTCAGAGGCACT		5.00 × 10^−6^	−3.53	50.2
PTEN9	Forward Primer	23	60	43.5	CTCTGTATTAGTGGCATCACAAG	417	6.00 × 10^−4^	−2.13	47.6
	Reverse Primer	23	60	43.5	CTCACTCGATAATCTGGATGACT		6.00 × 10^−4^	0.36	17.6
	Wild-Type Probe	30	68	43.5	TGACACCGCCAAATTTAATTGCAGAGGTAG		8.00 × 10^−8^	−0.31	29.8
	Mutant Probe	30	66	40	TGACACCGCCAAATTTAATTGCAGAGTTAG		1.00 × 10^−6^	−0.27	28.4
PTEN10 Wild-Type	Forward Primer	20	60	50	GCTACGACCCAGTTACCATA	172	0.021	0.37	17.9
	Reverse Primer	25	60	40	CACTGGTCTATAATCCAGATGATTC		5.00 × 10^−5^	−3.68	56.2
	Wild-Type Probe	30	68	53	CCACCAGGGAGTAACTATTCCCAGTCAGAG		8.00 × 10^−8^	−3.14	52.9
PTEN10 Mutant	Forward Primer	21	61	52.4	GCTACGACCCAGTTACCATAG	130	0.005	0.37	17.9
	Reverse Primer	22	60	40.9	ATAATACACATAGCGCCTCTGA		0.002	−0.14	27.4
	Mutant Probe	30	67	47	TTCTGTCCACCAAGGAGTAACTATTCCCAG		2.00 × 10^−5^	−1.28	49.3
PTEN11 Wild Type	Forward Primer	21	62	47.6	TTTCTGTCCACCAGGGAGTAA	148	0.005	−1.47	42.4
	Reverse Primer	21	62	47.6	CCGCCACTGAACATTGGAATA		0.005	−1.29	41.3
	Wild-Type Probe	24	68	54	TTCCCAGTCAGAGGCGCTATGTGT		2.00 × 10^−4^	0.22	22.2
PTEN11 Mutant	Forward Primer	20	60	50	GCTACGACCCAGTTACCATA	172	0.021	0.37	17.9
	Reverse Primer	25	60	40	CACTGGTCTATAATCCAGATGATTC		5.00 × 10^−5^	−3.68	56.2
	Mutant Probe	27	66	48	CTATTCCCAGTTAGAGGCGCTATGTGT		0.001	−0.19	28.4
PTEN12	Forward Primer	23	60	43.5	GGACGAACTGGTGTAATGATATG	153	6.00 × 10^−4^	0.13	21.9
	Reverse Primer	21	60	47.6	TCAGATCCAGGAAGAGGAAAG		0.005	−0.61	34.3
	Wild-Type Probe	30	66	40	TTAAAGGCACAAGAGGCCCTAGATTTCTAT		8.00 × 10^−8^	−2.28	40.4
	Mutant Probe	30	66	40	TAAAAGGCACAAGAGGCCCTAGATTTCTAT		1.00 × 10^−6^	−2.28	40.4
PTEN13	Forward Primer	24	61	41.7	GCTTGAGATCAAGATTGCAGATAC	202	2.00 × 10^−4^	−3.45	55.3
	Reverse Primer	19	61	52.6	TCGTGTGGGTCCTGAATTG		5.50 × 10^−2^	1.1	−2.5
	Wild-Type Probe	27	67	44	TAACCATGCAGATCCTCAGTTTGTGGT		4.00 × 10^−6^	−1.1	38.2
	Mutant Probe	27	68	48	ATGCAAATCCTCAGTTTGTGGTCTGCC		1.00 × 10^−3^	−1.76	40.2

Only one of the six TP53 mutations required separate primer pairs for the wild-type and mutant alleles, which was the TP53-4 mutation (Table 5). Primer lengths ranged from 17 to 23 bp, while amplicon lengths ranged from 83 to 272 bp. Primer Tms ranged from 60 to 63 °C, while GC contents ranged from 43.5 to 58.9%.

**Table 5 cimb-47-00292-t005:** Primers and probes designed to amplify and identify the selected TP53 gene mutations.

ID	OLIGO	Length	Tm	GC%	Sequence	Amplicon Length	BLAST e-Value	Hairpin ΔG	Hairpin Tm
TP53_1	Forward Primer	22	63	50	CAGTGTGATGATGGTGAGGAT	109	5.00 × 10^−3^	0.76	8.9
	Reverse Primer	22	62	50	TATCTCCTAGGTTGGCTCTGAC		2.00 × 10^−3^	−0.06	26
	Wild-Type Probe	24	69	54	CGCCCATGCAGGAACTGTTACACA		2.00 × 10^−4^	−2.72	57.3
	Mutant Probe	22	69	54	ATGCCGGCCATGCAGGAACTGTTA		0.042	−2.72	57.3
TP53_2	Forward Primer	22	62	50	CAGTGTGATGATGGTGAGGATG	127	0.002	0.76	8.9
	Reverse Primer	22	62	50	CTCATCTTGGGCCTGTGTTATC		0.002	0.87	7.8
	Wild-Type Probe	30	69	50	AGTTGTAGTGGATGGTGGTACAGTCAGAGC		8.00 × 10^−8^	−0.3	28.2
	Mutant Probe	30	69	50	TGGTTGTAGTGGATGGTGGTACAGTCAGAG		1.00 × 10^−6^	−0.3	28.2
TP53_3	Forward Primer	20	62	50	AGACTTGGCTGTCCCAGAAT	83	0.021	−1.18	40.7
	Reverse Primer	22	62	45.5	ATCTTCTGTCCCTTCCCAGAAA		0.002	−1.86	40.7
	Wild-Type Probe	24	69	58	ACGGTTTCCGTCTGGGCTTCTTGC (Anti-Sense)		2.00 × 10^−4^	−3.15	45.8
	Mutant Probe	24	69	58	ACGGTTTCCCTCTGGGCTTCTTGC (Anti-Sense)		0.042	−1.17	43.3
TP53_4 Wild-Type	Forward Primer	21	60	47.6	CATGAAGGCAGGATGAGAATG	184	0.005	−0.7	40.2
	Reverse Primer	18	60	50	TGAGCGCTTCGAGATGTT		0.22	−0.61	35.5
	Wild-Type Probe	26	68	54	AGTTCCAAGGCCTCATTCAGCTCTCG		1.00 × 10^−5^	−0.27	28.9
TP53_4 Mutant	Forward Primer	18	60	50	AAGGCCTCATTCAGCTCT	272	0.22	−0.27	28.9
	Reverse Primer	18	60	50	ATTGCACCATTGCACTCC		0.22	−2.45	51.2
	Mutant Probe	24	68	58	CAGAACATCTCGAAGCGCTCACGC		3.00 × 10^−3^	−2.96	61.9
TP53_5	Forward Primer	17	62	58.8	AGCCTGGGCATCCTTGA	273	0.86	−1.74	46
	Reverse Primer	19	61	58.9	CTGGGCAACAGAGTGAGAC		0.055	−1.69	49.4
	Wild-Type Probe	28	68	50	CCTCATTCAGCTCTCGGAACATCTCGAA		1.00 × 10^−6^	−0.57	33.6
	Mutant Probe	27	69	59	CAAACATCTCGAAGCGCTCACGCCCAC		7.00 × 10^−5^	−3.27	62.1
TP53_6	Forward Primer	23	61	43.5	GACTTAGTACCTGAAGGGTGAAA	106	6.00 × 10^−4^	−2.5	53.6
	Reverse Primer	21	61	47.6	CCTTTCCTTGCCTCTTTCCTA		5.00 × 10^−3^	3.78	−251.2
	Wild-Type Probe	30	68	50	TCTCCATCCAGTGGTTTCTTCTTTGGCTGG		8.00 × 10^−8^	−1.99	44.3
	Mutant Probe	30	68	50	ATCCATCCAGTGGTTTCTTCTTTGGCTGGG		3.00 × 10^−7^	−2.47	46.6

### 3.3. qPCR Probes

A wild-type and mutant probe was designed for each mutation. For all the mutations studies here, the Tms ranged from 64 to 70 °C, each probe being at least 5 degrees higher than the melting temperature of its primers (Table 2, Table 3, Table 4 and Table 5). Lengths of probes ranged from 22 to 30 bp, with GC contents all within 40–60%.

### 3.4. Primer/Probe Validation

Every oligonucleotide designed in this study was passed through BLAST to test for similarities to non-target regions of the human genome. Returned E-Values ranged from 0.021 to 8.00 × 10^−8^. However, several oligos designed for the PTEN mutations had shared homology with sequences from the PTENP1 gene located on chromosome 9. Therefore, it is recommended that these oligos be used with isolated chromosome 10 as the template DNA for the qPCR assay. The same can be performed with all mutations concerning their respective genes and chromosome locations.

The UCSC In Silico PCR tool was then used to ensure that the primer pairs only amplified their respective target regions. Aside from several PTEN gene mutations, all pairs achieved this due to the homologous PTENP1 gene located on chromosome 9. This again indicates the necessity of using isolated chromosome 10 as the template DNA for these mutations. Several PTEN mutations’ primer pairs also had matches to fix patch sequences, but these were the same target regions they were meant to amplify, found in a fix patch sequence for the primary genome assembly.

The Thermo Fisher Scientific Multiple Primer Analyzer was then used to check for self- and cross-dimerisation among primer pairs and their respective wild-type and mutant probes. Several oligos returned with an unwanted dimerisation level, so these were taken back and adjusted to avoid any dimerisation. Once this was achieved, the new oligos underwent the same validation processes as before until all fields were met with satisfactory results.

Each primer pair was checked to ensure that melting temperatures were within 1 °C of each other and that probe Tms were between 5 and 10 °C above those of their respective primers. When this was not the case, the oligos were redesigned to ensure correct melting temperatures. Every primer and probe designed was checked for hairpin formation, and we ensured that any secondary structure formed had a Tm lower than that of the primer/probe itself and that the ΔG value was less than values greater than −9.

## 4. Discussion

Over the years, there has been extensive research into how different mutations can affect an individual’s reaction to specific therapies, along with how different mutations can lead to more or less severe forms of diseases, including prostate cancer. The AR1 (rs137852578) mutation, also known as T878A, has been shown to enhance the agonistic effects of enzalutamide, thereby reducing the drug’s effectiveness [25,26]. However, Prekovic et al. [25] showed that abiraterone acetate could inhibit the cell proliferation of those cell lines containing the T878A mutation. AR2 (rs137852581), also known as H875Y, has also been found to be responsible for enzalutamide resistance by destabilising the binding of the drug [27]. Conteduca et al. [28] conducted a study where they found that the AR3 (rs864622007, or p.L702H) mutation was found in patients who had previously been treated with docetaxel but not in patients who had not received chemotherapy but had been treated with abiraterone. Altogether, this shows that the presence of these mutations could be brought about through chemotherapy agents such as docetaxel but can be avoided by using Androgen therapy agents like abiraterone acetate. However, if these mutations are present, the use of enzalutamide would be ineffective in the treatment of prostate cancer. Therefore, using these qPCR assays would be beneficial when deciding whether or not to use drugs such as enzalutamide or abiraterone to treat patients (Table 6).

A loss of the ATM protein through genetic mutations has been found in prostate cancer, with the loss resulting in genomic instability [10]. Kaur et al. [9] showed that ATM loss is associated with higher-grade tumours, showing a more severe form of prostate cancer. ATM1 (rs1565399936), having a stop-gained consequence, results in the protein encoding being cut short. With this mutation being in the first third of the protein’s total length, this would result in over two-thirds of the protein not being encoded, almost certainly resulting in a loss of function of this protein and, therefore, in genomic instability. The ATM2 and ATM5 (rs1438576066 and rs876660315, respectively) mutations have frameshift consequences, resulting in alterations to the subsequent protein sequence, thereby affecting the proper function of the protein, leading again to genomic instability. However, these effects can be treated with ATR inhibitors, but a combination of both ATR and PARP inhibitors is seen to work best [10,12], which shows that using these assays may help decide the best course of action in terms of treatments.

PTEN deletions are more common in prostate cancer than point mutations, with more point mutations being located within the phosphatase domain of the gene [29]. However, that is not to say that point mutations in this gene will not cause a loss of function, leading to the same outcomes as PTEN deletions. Less common methods of PTEN inactivation include missense mutations, while mutations such as stop-gained and splice variants may pose a more significant risk of PTEN deletion and inactivation, respectively [29,30]. PTEN2 (rs1114167621) is a splice acceptor variant located at the splice region between intron three and exon four of the PTEN gene. This mutation can severely alter the mRNA transcript for the gene, leading to incorrect protein coding by either including the intron or excluding the exon in the transcript [31]. The same is true for the two other splice acceptor variants used in this study, PTEN10 (rs786204862) and PTEN13 (rs876661024), which are located at the splice sites between intron five and exon six and intron six and exon seven, respectively. This exclusion of the exon, or the inclusion of the intronic sequence, will result in a protein sequence different to the expected PTEN protein, which can result in a non-functional protein, leading to prostate cancer progressing to severe forms of the disease [15,32]. PTEN9 (rs587776667) is a splice donor variant located at the splice region between exon four and intron four. This mutation can lead to intronic sequences being included in the final mRNA transcript [31]. This, again, results in a mutant protein sequence that will not function as it should, leading to advanced prostate cancer. These assays could prove useful as part of a primary diagnosis, packaged with more common tests such as PSA tests.

Four of the PTEN mutations studied here had stop-gained consequences, those being PTEN4, PTEN5, PTEN11, and PTEN12 (rs121909219, rs121909224, rs786204864, and rs786204933, respectively). PTEN4 is located within the first half of exon seven. A stop-gained mutation in this codon would result in the rest of this and the remaining two exons not being encoded, leading to a shortened protein. This mutation falls within the C2 domain of the protein and would cut out the entire C-tail [33]. The C-tail acts as a regulator for the protein, and so its loss results in PTEN inactivation, ultimately leading to prostate cancer progression [34]. PTEN5 and PTEN12 are located within exon 5. In contrast, PTEN11 is situated in the sixth exon, meaning that the stop-gained mutations would result in the entire C-terminal region of the protein not being encoded. Again, these assay can be beneficial in treatment monitoring due to PTEN’s common association with increased mutations in developing prostate cancers.

According to Álvarez-Garcia et al. [35], less than 5% of initial-stage prostate cancers contain missense mutations, while almost 70% of metastatic forms of the disease contain these mutations. Therefore, it can be deduced that testing for the presence of these forms of mutation can indicate progressive forms of prostate cancer. All five missense mutations studied here are located within sections of the gene that encode the PTEN protein’s phosphatase region. PTEN’s role in regular cell function is to regulate the PI3K signalling cascade by dephosphorylating PIP3 into PIP2, acting as an antagonist to the pathway [33]. The protein’s phosphatase region is directly responsible for this dephosphorylation. Therefore, any mutations that alter this region can have dire consequences for the functioning of this region.

As PTEN acts as a negative regulator of the PI3K pathway, which is directly linked to cell growth, proliferation, and survival regulation, the protein plays a vital role in ensuring cancer cells do not survive [36]. Therefore, one treatment method for PTEN loss would be to use therapies that mimic its effects. Several therapies have been, or are currently being, tested for use in treating PTEN loss, which inhibits the PI3K/AKT/mTOR pathway. These include mTOR inhibitors [37], PI3K inhibitors [38,39], and AKT inhibitors [40]. When used with other therapies, such as taxane chemotherapeutics and PARP inhibitors, these treatments can have meaningful results on individual patient outcomes, showing that determining PTEN loss can allow for better treatment options. Along with their uses, these PTEN assays could be potential indicators of personalised treatment options involving PI3K inhibitors.

The TP53 and its encoded protein, p53, have been associated with several different cancers over the years, including breast cancer, head and neck cancer, lung cancer, and prostate cancer [41,42,43,44]. The presence of TP53 missense mutations is often associated with p53 nuclear accumulation, which itself is found to increase tumour-infiltrating lymphocytes [45]. However, this is to be expected, considering that TP53 missense mutations are associated with genomic instability, leading to a greater immune response [46]. There have been several studies that have shown significant improvements in prostate cancer patients treated with immunotherapies [47,48,49]. TP53-1 (rs985033810), TP53-2 (rs587782289), and TP53-3 (rs11540654) are all missense mutations, showing a possible target for immunotherapy treatments. The inactivation of the TP53 is found in most prostate cancers with TP53 mutations, making up over 90% of all these cases [50]. This inactivation can be achieved through missense mutations, which make up most cases, but also through frameshift and stop-gained mutations [51]. Therefore, TP53-4 (rs730882029) and TP53-6 (rs876659384), being stop-gained mutations, and TP53-5 (rs1131691022), being a frameshift mutation, show their potential as markers of TP53 inactivation in prostate cancer, showing potential for immunotherapy treatments.

Although the development of qPCR or RT-qPCR assays has been performed for prostate cancer uses, this study’s assays differ in uses. For example, the TP53 qPCR assays designed here may be used when determining whether immunotherapy approaches will be effective, while a previous assay by Gonen-Korkmaz et al. [52] was used to determine whether treatment with TNF-α was effective after the fact. The AR assays in the current study can be used to indicate whether certain drugs will be effective treatment options, but Latil et al. [53] developed AR assays to determine AR expression levels to determine whether treatments already used have worked. Uygur et al. [54] developed a qPCR assay to determine PTEN expression levels and whether treatment with SLUG inhibitors, a zinc transcription factor, effectively elevated the PTEN expression levels back to normal. In contrast, the PTEN qPCR assays here are directed more towards diagnosis and personalised treatment but may also be used for treatment monitoring.

As this study was purely an in silico-based design, the next step in this research would be to perform in vitro tests to confirm the primers and probes’ performance in the real world. For this next research phase, the first step would be synthesising the primers and probes and performing a no-template control to ensure no primer dimer formation by examining the melt curves. The next step would be to use template DNA to ensure amplification efficiency and specificity while simultaneously testing probe fluorescence in positive templates and no fluorescence in no-template controls/non-target DNA. The final step would then be testing the reproducibility of results.

## 5. Conclusions

Knowing that prostate cancer is one of the most common causes of cancer-related death among men worldwide, it is of the utmost importance that research be performed to improve the outcomes of all patients suffering from this disease. Suppose there are methods to identify key mutations that may trigger specific responses to certain treatments or show signs of cancer progression. In that case, these should be studied further to allow the development of assays or tests that progress the fight against prostate cancer. The five genes studied here are all commonly associated with prostate cancer, with a special focus on the AR and PTEN genes, as these two genes are the most frequently altered in this disease. The qPCR probes designed in this study could help many patients choose the best treatment possible for their cancer cases. These qPCR assays need further experimental validation before clinical use, but they are a start towards a better future in the treatment of prostate cancer.

## Figures and Tables

**Figure 1 cimb-47-00292-f001:**
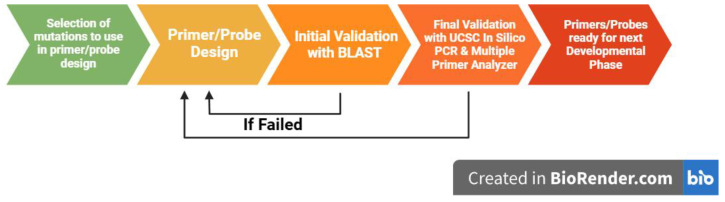
Workflow diagram outlining the steps taken in this study from mutation selection to final primer/probe validation. Figure created with BioRender (BioRender.com).

**Table 1 cimb-47-00292-t001:** A list of the mutations used in the current study for qPCR probe design.

Gene	Mutation ID	Consequence	Chromosome	Position	Reference Allele	Alternative Allele	Code
AR	rs137852578	Missense	X	67723710	A	G	AR1
	rs137852581	Missense	X	67723701	C	T	AR2
	rs864622007	Missense	X	67711621	T	A	AR3
	rs2147530924	Stop-Gained	X	67717559	G	A	AR4
ATM	rs1565399936	Stop-Gained	11	108259026	T	G	ATM1
	rs1438576066	Frameshift	11	108315831	C	CC	ATM2
	rs1555084931	Splice Acceptor Variant	11	108271250	G	A	ATM3
	rs747727055	Missense	11	108245000	C	T	ATM4
	rs876660315	Frameshift	11	108244097	C	-	ATM5
PTEN	rs1085308046	Missense	10	87933160	T	G	PTEN1
	rs1114167621	Splice Acceptor Variant	10	87931045	G	A	PTEN2
	rs1114167645	Missense	10	87933143	G	C	PTEN3
	rs121909219	Stop-Gained	10	87957915	C	T	PTEN4
	rs121909224	Stop-Gained	10	87933147	C	T	PTEN5
	rs121909229	Missense	10	87933148	G	A	PTEN6
	rs121913293	Missense	10	87952142	C	T	PTEN7
	rs121913294	Missense	10	87952143	G	A	PTEN8
	rs587776667	Splice Donor Variant	10	87931090	G	T	PTEN9
	rs786204862	Splice Acceptor Variant	10	87952117	G	A	PTEN10
	rs786204864	Stop-Gained	10	87952136	C	T	PTEN11
	rs786204933	Stop-Gained	10	87933196	T	A	PTEN12
	rs876661024	Splice Acceptor Variant	10	87957852	G	A	PTEN13
TP53	rs985033810	Missense	17	7674232	C	G	TP53-1
	rs587782289	Missense	17	7674257	A	G	TP53-2
	rs11540654	Missense	17	7676040	C	G	TP53-3
	rs730882029	Stop-Gained	17	7670685	G	A	TP53-4
	rs1131691022	Frameshift	17	7670686	GA	A	TP53-5
	rs876659384	Stop-Gained	17	7673552	C	A	TP53-6

**Table 6 cimb-47-00292-t006:** A brief overview of the possible uses of the assays designed in the current study.

Gene	Assay Uses
AR	Personalised treatments involving enzalutamide/arbiraterone acetate
ATM	Personalised treatments involving ATR inhibitors
PTEN	Primary diagnosis; treatment monitoring; personalised treatments involving PI3K inhibitors
TP53	Personalised treatments involving immunotherapy

## Data Availability

The original contributions presented in this study are included in the article. Further inquiries can be directed to the corresponding author(s).

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
