# Peer review of "In Silico Design of Quantitative Polymerase Chain Reaction (qPCR) Assay Probes for Prostate Cancer Diagnosis, Prognosis, and Personalised Treatment"

_cimb, 2025, doi:10.3390/cimb47040292_

Round 1
Reviewer 1 Report
Comments and Suggestions for Authors
I reviewed the manuscript by Trevor Kenneth Wilson and Oliver Tendayi Zishiri, which presents the in silico design of qPCR assay probes targeting key genes associated with PCa (AR, ATM, PTEN, TP53). The study addresses an important topic in molecular diagnostics and proposes initial designs for future validation. Overall, I consider the paper is well organized and presents interesting preliminary data. Nevertheless, several aspects could be improved to enhance the scientific robustness and future applicability of the findings.
Comments
-The study would benefit from a more detailed discussion of the next steps required to validate the designed assays. It would be valuable to outline a future validation plan including experimental testing of the probes' performance, specificity, and amplification efficiency in vitro.
-Although the in silico design is carefully described, it would strengthen the ms if the authors could include predictive modeling of probe performance (for example, predicted melting temperatures, secondary structure formation, dimerization risk). Were such in silico analyses performed? If so, please include them; if not, it would be useful to suggest them as part of future optimization.
-The discussion could be expanded by comparing the proposed assays with existing qPCR designs reported for PCa biomarkers. How do the sequences and targets designed here differ or improve upon previous studies? Including a comparative table would significantly increase the impact of the review.
-The clinical relevance of selecting these four genes is clear, but the authors could elaborate on the specific clinical contexts where their assays could be applied (e.g., diagnosis, risk stratification, treatment selection). Are the designed assays intended for primary diagnosis, treatment monitoring, or personalized medicine applications? Not clear
-Figures and tables could be improved for clarity. For instance, a workflow diagram summarizing the bioinformatics pipeline (from gene selection to probe design and validation) would make the methodology more transparent and easy to follow-
-The results section could also benefit from a more quantitative evaluation of the designed primers/probes, such as GC content, amplicon length, and predicted efficiency scores.
Minor comments
As a minor point, it would improve the Introduction if the authors briefly mention the epidemiology of prostate cancer in Africa, to better contextualize the relevance and potential regional impact of their study.
In the abstract, specifying that the study is an in silico design would immediately set accurate expectations for readers.
Some sections of the manuscript would benefit from minor language revisions to improve readability. For example, the introduction could be more concise and the transition between paragraphs made smoother.
Standardize the reference style throughout the manuscript. Some citations appear inconsistent.
In the M&M section, more precise details about the software settings (e.g., thresholds for primer specificity during BLAST) would improve reproducibility.
Comments on the Quality of English LanguageSome sections of the manuscript would benefit from minor language revisions to improve readability.
Reviewer 2 Report
Comments and Suggestions for Authors
Wilson et al. have submitted a manuscript with title: “In silico design of Quantitative Polymerase Chain Reaction (qPCR) assay probes for prostate cancer diagnosis, prognosis, and personalised treatment“.
Overall, the presentation quality is high, text is well written, direct to the goal, and easy to follow. Also, the methodology well explained and the references appropiated.
However, more than a research article, I find it a “study design”. No experiments were performed, no real results are presented here. Have the authors already tested the primers in patient samples?
From my side, the manuscript reaches the quality necessary for publication, but it is the editorial team who should decide if they want to publish a “in silico design” as an article or better in another publication format.
Minor comments:
In case of acceptance for publication by the editorial team, I recommend to add an additional table summarizing the recommended therapy based in the mutations found.
